# Identification of Genes Associated with Prognosis and Immunotherapy Prediction in Triple-Negative Breast Cancer via M1/M2 Macrophage Ratio

**DOI:** 10.3390/medicina59071285

**Published:** 2023-07-11

**Authors:** Jianyu Liu, Yuhan Deng, Zhuolin Liu, Xue Li, Mingxuan Zhang, Xin Yu, Tong Liu, Kexin Chen, Zhigao Li

**Affiliations:** Department of Breast Surgery, Harbin Medical University Cancer Hospital, No. 150 Haping Road, Nangang District, Harbin 150081, China; jyls865184@hrbmu.edu.cn (J.L.); 2020021711@hrbmu.edu.cn (Y.D.); 2021021842@hrbmu.edu.cn (Z.L.); euxil@hrbmu.edu.cn (X.L.); 2021021884@hrbmu.edu.cn (M.Z.);

**Keywords:** TAMs, TNBC microenvironment, immune infiltration, immunotherapy, riskScore model

## Abstract

*Background and Objectives:* Triple-negative breast cancer (TNBC), a highly aggressive and heterogeneous subtype of breast cancer, accounts for ap-proximately 10–15% of all breast cancer cases. Currently, there is no effective therapeutic target for TNBC. Tu-mor-associated macrophages (TAMs), which can be phenotypically classified into M1 and M2 subtypes, have been shown to influence the prognosis of various cancers, including ovarian cancer. This study aimed to investigate the role of M1/M2 macrophages in the TNBC tumor microenvironment (TME), with a focus on identifying prognostic genes and predicting immunotherapy response. *Materials and Methods:* The study employed the CIBERSORT algorithm to analyze immune cell expression in the TME. Genes associated with the M1/M2 macrophage ratio were identified using Pearson correlation analysis and used to classify patients into dis-tinct clusters. Dimensionality reduction techniques, including univariate Cox regression and Lasso, were applied to these genes. The expression of prognostic genes was validated through immunohistochemistry. *Results:* The study found a high prevalence of TAMs in the TME. Among the patient clusters, 109 differentially expressed genes (DEGs) were identified. Three significant DEGs (LAMP3, GZMB, and CXCL13) were used to construct the riskScores. The riskScore model effectively stratified patients based on mortality risk. Gene Set Enrichment Analysis (GSEA) associated the riskScore with several significant pathways, including mismatch repair, JAK/STAT3 signaling, VEGF signaling, antigen processing presentation, ERBB signaling, and P53 signaling. The study also predicted patient sensitivity to im-munotherapy using the riskScores. The expression of the three significant DEGs was validated through immunohisto-chemistry. *Conclusions:* The study concluded that the riskScore model, based on the M1/M2 macrophage ratio, is a valid prognostic tool for TNBC. The findings underscore the importance of the TME in TNBC progression and prognosis and highlight the po-tential of the riskScore model in predicting immunotherapy response in TNBC patients.

## 1. Introduction

Breast cancer (BC) is the most prevalent malignancy among women worldwide and is a leading cause of cancer-associated mortality. In 2020, an estimated 226,419 new cases of breast cancer were diagnosed, resulting in 684,996 deaths globally, according to the Global Cancer Observatory (GCO). Triple-negative breast cancer (TNBC) is defined as breast cancer that lacks the expression of an estrogen receptor (ER), progesterone receptor (PR), and human epidermal growth factor receptor 2 (HER2). This subtype of breast cancer presents significant therapeutic challenges since most drug treatments target only one of these three receptors [1]. Given the complex interplay between tumor and non-tumor cells and biomolecules in the tumor microenvironment (TME), the TME has emerged as an attractive target for TNBC therapy [2]. Tumor-associated macrophages (TAMs) are a crucial component of the TME and are therefore of particular interest in the treatment of TNBC.

A significant proportion of TAMs present in TME are derived from circulating monocytes [3], which differentiate into non-polarized (M0) macrophages upon stimulation with monocyte colony-stimulating factor. M0 macrophages are responsive to environmental cues and undergo phenotypic polarization, resulting in distinct M1 and M2 macrophage populations. M1 macrophages, also known as classically activated macrophages, are induced by Type 1 T helper cell (Th1) cytokines such as interferon interferon-γ (IFN-γ) or tumor necrosis factor (TNF) [4]. These macrophages exhibit potent anti-tumor effects by releasing pro-inflammatory cytokines (such as TNF and interleukin (IL)-2) and active nitrogen and oxygen intermediates [5]. In contrast, M2 macrophages are stimulated by Th2-type T helper cytokines (such as IL-4, IL-10, and IL-13) and exhibit a tumor-promoting profile [5]. The majority of TAMs within the TME are associated with the M2 macrophage phenotype [6]. A meta-analysis of over 2000 breast cancer patients revealed that high densities of TAMs within the TME were predictive of a worse prognosis [7]. However, the polarization of TAMs and their internal gene regulation may be more critical than their mere presence. For instance, the M1/M2 macrophage ratio could potentially impact breast cancer prognosis, and the genes that influence this ratio may also play a role in patient prognosis, as has been demonstrated in ovarian cancer [8] In this study, we used a computational algorithm to estimate TAMs based on clinically annotated TNBC gene expression profiles. We estimated the infiltration pattern of the M1/M2 macrophage ratio in tumor samples from 219 TNBC patients from the TCGA and GEO datasets and systematically correlated this ratio with the genomic features and clinical characteristics of TNBC. Ultimately, we developed a model to quantify the infiltration pattern of M1/M2 macrophages (riskScore) and found that riskScore is a robust prognostic biomarker and predictor of treatment response to immune checkpoint inhibitors (ICIs).

## 2. Materials and Methods

### 2.1. TNBC Dataset and Pretreatment

We systematically searched for TNBC gene expression datasets that were publicly available and reported complete clinical annotations. A total of 219 samples were collected from the TCGA and GEO databases. For the TCGA cohort (114 samples), RNA-seq data and corresponding clinical information were retrieved from the TCGA database (http://cancergenome.nih.gov/ (accessed on 3 January 2021)), and count values were converted into TPM values that were more similar to microarray results, making them more comparable between samples [9].The microarray dataset of the GSE58812 cohort generated by the Affymetrix Human Genome U133 Plus 2.0 Array, as well as the corresponding clinical data from 105 samples (excluding those with a survival datum of 0), were downloaded from the GEO website (https://www.ncbi.nlm.nih.gov/geo/ (accessed on 3 January 2021)). The raw data of the dataset were then preprocessed using the RMA background adjustment algorithm [9]. The data were analyzed using (R (version 3.6.1): R Foundation, Vienna, Austria) and the R Bioconductor package (Fred Hutchinson Cancer Research Center, Seattle, WA, USA).

### 2.2. Identification of M1/M2 Macrophage-Related Genes and Clustering

Using R, version 3.6.1, a Pearson correlation analysis was employed to screen the genes most associated with the M1/M2 macrophage ratio (*p*-value < 0.01), and the relationships between individual genes and the M1/M2 macrophage ratio were quantified via the Pearson correlation coefficient (r-value). A consensus clustering algorithm for the M1/M2 macrophages correlation gene analysis (30) was used to classify patients into M1M2 clustering groups. A principal component analysis (PCA) was used to validate the success of the clustering. DEGs between groups were identified using the R package limma [10], which uses an empirical Bayesian approach to estimate gene expression changes using a conditioning *t*-test. The selected DEGs were all statistically significant (corrected *p*-value < 0.05). A univariate Cox regression and Lasso were used for dimensionality reduction to reduce noise or redundant genes. The final obtained genes were used to build the riskScore model.

### 2.3. Inference of Immune Infiltration Microenvironment

To quantify the relative proportions of 22 immune cell types in tumor tissue, we used the CIBERSORT algorithm [11], which allows for the sensitive and specific differentiation of immune cell phenotypes, including B cells, T cells, NK cells, macrophages, dendritic cells (DCs), and myeloid subpopulations. The ESTIMATE algorithm was applied to the levels of the ImmuneScore and StromalScore of each sample. MCpounter, ssGSEA, and TIMER were also used to verify that there was significant infiltration of immune cells into the TME.

### 2.4. Genomic Alterations in Grouping

Xenahubs (https://gdc.xenahubs.net/download/TCGA-BRCA/Xena_Matrices/TCGA-BRCA.mutect2_snv.tsv.gz, (accessed on 3 January 2021) was used to download somatic mutations, including single-nucleotide polymorphisms (SNPs) or single-nucleotide variants (SNVs), and copy number variations (CNVs) that corresponded to the RNA-seq data. A GISTIC analysis was employed to determine genomic enrichment, and GISTIC 2.0 (https://gatk.broadinstitute.org (accessed on 3 January 2021) was used to obtain copy number thresholds for CNV and mutation peaks in both groups.

### 2.5. Functional and Pathway Enrichment Analysis

A gene annotation enrichment analysis of DEGs was carried out using the clusterProfiler R package (Fred Hutchinson Cancer Research Center, Seattle, WA, USA) [12]. According to GSEA7.2, the expression of the GO gene set and KEGG pathway was up-regulated or down-regulated in high and low riskScore populations (*p*-value < 0.01).

### 2.6. Prediction of Immunotherapy Response

The IMvigor210 cohort is a cohort of uroepithelial cancers treated with the anti-PD-L1 antibody Atezolizumab to predict patient response to immunotherapy [13]. Based on the Creative Commons 3.0 license from (http://research-pub (accessed on 10 January 2021)), we downloaded the full expression data and clinical data. The raw data were then normalized using the DEseq2 R package (Bioconductor project, Fred Hutchinson Cancer Research Center, Seattle, WA, USA), and the count values were converted into TPM values.

### 2.7. Prediction of Immunotherapy Sensitivity

Data on the immunotherapy sensitivity of TCGA-BRCA patients were extracted from The Cancer Immunome Atlas database (https://tcia.at/patients (accessed on 20 January 2021)). The database can query data on the gene expression of specific immune-related gene sets, the cell compositions of immune infiltrates, and tumor heterogeneity. According to the sensitivity scores of TCGA-BRCA patients to PD-L1 and CTLA4 inhibitors in the TCIA database, the differences in sensitivity to immunotherapy between the groups with high and low riskScores were analyzed.

### 2.8. Immunohistochemistry

Immunohistochemistry (IHC) was used to validate the protein expression of prognostic genes in TNBC tissue. Thirty TNBC tissue samples were stained using anti-LAMP3, anti-GZMB, and anti-CXCL13 antibodies. The tissue samples were fixed in 10% formalin at room temperature, embedded in paraffin, and processed into 4mm continuous sections. In brief, the tissue sections were deparaffinized, and antigen retrieval was carried out via boiling for 10 min in 10 mmol/L of citrate buffer (pH = 6.4). Subsequently, sections were treated with 3% hydrogen peroxide in methanol to inactivate endogenous peroxidase and treated with citrate buffer (pH = 6.0) for optimal antigen retrieval. To block non-specific binding, sections were incubated with 1% bovine serum albumin in phosphate-buffered saline for 30 min. Primary antibody staining was performed, and sections were incubated overnight at 4 °C. Thereafter, the sections were gently washed three times for 5 min in phosphate-buffered saline, and control staining was performed using secondary antibody only to ensure specificity. Finally, the samples were sealed, observed, and photographed using an optical microscope. The staining score was independently evaluated by assessing the integrated staining intensity and the proportion of positive cells. The final score was determined by adding the average proportion of staining intensity scores and the proportion of positive cell scores, with scores ranging from 0 to 7. Samples with scores between 0 and 3 were defined as low expression, while those with scores between 4 and 7 were defined as high expression.

### 2.9. Survival Analysis

A Kaplan–Meier survival analysis was conducted on patients with differential protein expression or patients in high-/low-risk groups to perform survival analyses for individual proteins. Patients were divided into high/low expression groups based on median protein expression levels. To analyze the survival of the three combined proteins, the patients in the TCGA dataset were divided into high-/low-risk groups based on the Cox regression model. For a survival analysis of the combined 3 proteins, clinical patients with high levels of expression of more than 2 proteins were classified into the low-risk group and the remaining patients were classified into the high-risk group. Statistical significance was assessed using the log-rank test (*p*-value < 0.05).

## 3. Results

### 3.1. Landscape of the Tumor Immune Microenvironment in TNBC

The flow chart of this study is shown in Appendix A. Firstly, we combined the expression data converted from TCGA into TPM with the GEO data, adjusted with the RMA algorithm. Detailed clinical and pathologic characteristics of the patients are included in the information in Appendix A. The CIBERSORT algorithm was used to display the expression levels of several types of immune cells, such as M0/1/2 macrophages, DCs, and neutrophils. The results showed that TAMs were present in large numbers, accounting for more than 50% of TME cells (Figure 1A). Next, we explored the relationship between TAMs and TNBC survival (Figure 1B). The samples were grouped by median M1 macrophages and M2 macrophages in the TME. Both M1 and M2 macrophages were significantly correlated with survival (*p*-value < 0.05). The presence of M1 macrophages was associated with a favorable prognostic factor for survival, while the presence of M2 macrophages was considered a negative prognostic factor (Figure 1C). We also found a negative correlation between M1 macrophages and regulatory T cells (Tregs). This observation is consistent with previous studies, indicating that M1 macrophages can suppress Tregs through direct contact [14] or by inhibiting their accumulation in the tumor microenvironment (TME) via the secretion of soluble factors such as tumor necrosis factor (TNF) [15]. Notably, the TNF produced by M1 macrophages has been shown to reduce the inhibitory activity of Tregs via the NF-κB pathway. In contrast, we observed a positive correlation between M2 macrophages and monocytes. This finding aligns with the existing literature, suggesting that the majority of tumor-associated macrophages (TAMs) in the TME exhibit characteristics similar to the M2 macrophage phenotype [6]. This revision aims to simplify the language and improve the flow of ideas, which should make the paragraph easier to understand. However, the specific content and references have been preserved.

### 3.2. Determining the M1/M2 Macrophage Ratio as a Potential Prognostic Marker

Next, we proceeded to determine the prognostic value of the M1/M2 macrophage ratio for TNBC. Patients were divided into two groups based on the median M1/M2 macrophage ratio, and a survival analysis revealed significant differences between those with high and low ratios. To determine the potential prognostic value of the M1/M2 macrophage ratio, we performed a Pearson correlation analysis of genes related to the ratio. A Pearson correlation coefficient threshold of |r| > 0.3 was used to select the 425 most relevant genes. Subsequently, 191 genes associated with survival were extracted from M1/M2 macrophage genes via univariate Cox regression for dimensionality reduction (*p*-value < 0.05).

To better investigate the prognostic value of the M1/M2 macrophages, we used a consistent clustering algorithm to cluster a sample based on 191 genes. The optimal number of clusters was evaluated by the ConsensusClusterPlus software package. The clustering results were most stable when the number of individuals was two (K = 2) (Appendix A). The groups defined based on the 191 genes associated with survival are called M1M2cluster1 and M1M2cluster2, and they showed different genomic patterns (Figure 2A). A survival analysis of the two clusters confirmed that M1M2cluster1 had a significantly lower survival curve than M1M2cluster2 (Figure 2B,C). A PCA analysis also showed that we successfully differentiated the sample into two clusters (Figure 2D).

### 3.3. Construction of the riskScore Model and Its Functional Annotation

To identify potential biological differences between M1M2cluster1 and M1M2cluster2, we determined 109 DEGs between the two clusters using the R package limma (Bioconductor project, Fred Hutchinson Cancer Research Center, Seattle, WA, USA). A GO functional enrichment analysis revealed that these genes are concentrated in pathways involved in cytokine receptor activity and T cell activation. A KEGG analysis revealed that the genes are in the antigen-processing presentation pathway. A KEGG analysis showed that the genes were enriched in the antigen-processing presentation pathway (Figure 3A). The three most significant DEGs (LAMP3, GZMB, and CXCL13) were obtained via a Cox regression analysis and Lasso regression analysis: patients with higher riskScore values tended to express low levels of LAMP3, GZMB, and CXCL13 (Figure 3B). In the TGCA and GEO datasets, respectively, a survival analysis showed that patients with different mortality risks could be separated well using by high and low riskScore groups (Figure 3C,D). In addition, the sanger plot showed considerable agreement between M1M2 clustering and riskScore (Figure 3E). To further test the validity of the riskScore model, we performed a ROC analysis with a one-year area under the curve (AUC) of 0.638, indicating a relatively low prognostic value. However, the AUCs for the three-year and five-year ROC analyses were 0.724 and 0.713, respectively, confirming that riskScore is a valid prognostic marker for predicting the survival status of TNBC patients at three and five years (Figure 3F). 

Considering the importance of these three genes, we conducted a comprehensive analysis of the clinicopathological correlations of the LAMP3, GZMB, and CXCL13 proteins, including a survival analysis (Appendix A) and their expression levels across different T stages, N stages, and overall stages (Appendix A). Our findings revealed that survival rates did not significantly differ when groups were formed solely based on the cutoff values of these three proteins. Furthermore, there were no significant differences in the expression levels of these proteins across the two M stages and four T stages. However, a statistically significant difference was observed in the expression of all three proteins between the N1 and N3 stages and between the N1 and N2 stages for GZMB and CXCL13. These results provide valuable insights into the clinicopathological significance of the LAMP3, GZMB, and CXCL13 proteins in the context of the disease under study.

In addition, we conducted a Spearman correlation analysis to investigate the relationship between the expression levels of LAMP3, GZMB, and CXCL13 proteins and the cellular composition of the tumor microenvironment. Our analysis revealed a positive correlation between the expression levels of these proteins and the presence of exhausted T cells and a negative correlation with CD8 naive T cells (Appendix A).

### 3.4. riskScore Model-Related GSEA Enrichment Pathways

To gain insight into the relationship between genomic features and riskScore values, we applied a GSEA to interpret the gene expression data from the TCGA and GEO datasets. The set of genes associated with macrophage activation and migration, GM-CSF production, and monocyte differentiation to TAMs was relatively inactive in samples with higher riskScore values (Appendix A). In contrast, an ssGSEA showed sparse infiltration of monocytes, Tregs, and macrophages into the TME (Figure 4). Higher riskScore values were associated with an increased abundance of M2 macrophages within the TME, suggesting that our riskScore model predominantly influenced the transformation of M0 to M2 macrophages rather than the recruitment of monocytes or macrophages into the TME. Previous studies have explored the use of selective monocyte-targeted chemotherapeutic agents, such as trabectedin or CSF1 inhibitors, to inhibit TAMs and reduce their infiltration in BC xenograft mouse models, leading to prolonged survival. However, our GSEA results indicate that a higher riskScore value may serve as an unfavorable predictor for the efficacy of such treatments [16,17]. Thus, our study highlights the potential clinical implications of targeting the M1/M2 macrophage balance in TNBC and underscores the importance of incorporating a riskScore assessment into the development of personalized treatment strategies.

### 3.5. Correlation Study of TME Characteristics, Clinical Characteristics, and riskScore

We later focused on the correlations between TME characteristics and riskScore values. The correlations between expression levels of different cell types and riskScore were comprehensively evaluated (Figure 4). The CIBERSORT algorithm showed that the expression of several types of immune cells, including monocytes and M0/2 macrophages, increased with increasing riskScore values, while M1 macrophages decreased with increasing riskScore values, corresponding to the consistency of the M1/M2 macrophage ratio with riskScore values (Figure 5). Similarly, MCPcounter, ssGSEA, and TIMER analyses all showed low levels of infiltration of immune cells in the TME. Correspondingly, the KEGG pathways associated with cell proliferation, cell cycle, and antigen presentation were all negatively correlated with the riskScore (Figure 5). An exception was the number of fibroblasts, which increased with a higher riskScore value, consistent with several examples of active GO gene sets in high-riskScore samples, such as the fibroblast growth factor receptor signaling pathway (Appendix A). In addition, we examined the association between the ESTIMATE score (an indicator of tumor biological behavior) of the immune infiltrative microenvironment and the riskScore as well as the level of immune cells in TCGA and GEO. The results of ESTIMATEScore, ImuneScore, and StromalScore were higher when riskScore was lower, suggesting that riskScore plays an important role in predicting the anti-tumor immune microenvironment (Figure 5).

Significant differences were observed in the AJCC staging and N grade of TNBC, with higher riskScore values being associated with later staging and a higher N grade. However, no statistically significant differences were observed in age, metastasis, or T grade with respect to riskScore (Appendix A). These clinical findings indicate that high riskScore values are indicative of greater propensities for malignancy in TNBC patients.

### 3.6. The High riskScore Group Showed More Malignant Genomic Features

Genomic traits in high and low riskScore groups were studied using the TCGA dataset for somatic mutation analysis and CNVs. A global CNV map was drawn by comparing two clusters (Figure 6). By analyzing the mutation annotation file of the TCGA-BRCA cohort, the top 22 genes with the most tumor-associated mutations were listed (Figure 6A). A somatic mutation analysis showed that the TP53 (85%), TTN (21%), KMT2D (13%), and DYNC2H1 (11%) mutations were most enriched in the high riskScore group. TP53 (86%), TTN (29%), FAT3 (25%), and MUC16 (21%) mutations were significantly enriched in the low riskScore group (Figure 6B,C). Preclinical [18] and clinical [19] reports have detailed relationships between individual mutated genes and ICI response or resistance. However, relatively few genes in the TCGA-BRCA family are fully associated with sensitivity or resistance, such as PI3KCA and KMT2D. Given the function and high mutation rate of KMT2D in high riskScore groups, we will focus on this gene. A study by Wang et al. [20] showed that KMT2D mutant tumors in mice and humans are characterized by an increased infiltration of immune cells such as macrophages with cytotoxic T cells. Furthermore, using CRISPR-mediated genetically engineered mouse models (CRISPRGEMMs) [20], Wang et al. found that KMT2D deficiency acts as an independent factor that sensitizes tumors to responses to ICIs for multiple cancer types. In summary, further studies are needed to explore individual mutations and their roles in cancer immunity and immunotherapy and then use the results obtained to establish appropriate strategies to select suitable patients to receive ICIs in TNBC.

### 3.7. riskScore Group Predicts Response to Immunotherapy

ICIs are promising agents that block the binding of checkpoint proteins such as PD-1, CTLA-4, and ICOS to their receptor proteins. The use of ICIs has emerged as an anti-cancer treatment with unprecedented synergistic survival benefits [21,22]. Therefore, we next explored the predictive and prognostic value of the riskScore model for ICIs. We found that a high riskScore value was significantly associated with multiple immune checkpoint molecules, including PD-1, PD-L1, ICOS, and CTLA4 (Figure 7A–D). In addition, the ImmunophenotypeScore (IPS) was higher in the high riskScore group (Figure 7E). When further stratifying the immunotherapy response (including the complete response (CR), progressive disease (PD), partial response (PR), and stable disease (SD) groups) (Figure 7F), we found that the riskScore model could be a useful tool in predicting the clinical benefit of TNBC patients receiving immunotherapy, as patients with higher riskScore values were found to have significantly lower response rates to ICIs. While TMB and MSI are important factors affecting the efficacy of immunotherapy, our study did not find a significant association between riskScore values and TMB or MSI. Therefore, riskScore may be a more reliable predictor of clinical response to ICIs in TNBC patients compared to TMB or MSI. These results highlight the potential clinical utility of riskScore as a prognostic biomarker for TNBC patients receiving immunotherapy (Figure 7G).

To validate this result, we conducted additional analyses focusing on the prediction of immunotherapy response. Specifically, we compared the expression levels of immune checkpoints between high and low riskScore groups. Our results showed significant differences in the expression of these immune checkpoints, suggesting that the riskScore model may have potential utility in predicting the response to immunotherapy. Furthermore, we evaluated The Cancer Immunome Atlas (TCIA) scores for patients in the high and low riskScore groups (Appendix A). The TIDE score is a computational method that predicts the response to immune checkpoint blockade therapy. Interestingly, we found that patients in the low riskScore group had lower TIDE scores, indicating higher likelihoods of responding to immune checkpoint inhibitors.

### 3.8. riskScore Model Validation in Clinical Sample Tissues

To confirm the reliability of the identified DEGs (CXCL13, LAMP3, and GZMB), we used IHC to detect the protein expression of three genes in 30 clinical TNBC tissues (Figure 8A). We demonstrated that the mRNA levels of CXCL13, LAMP3, and GZMB significantly correlated with survival times in the TCGA and GEO datasets. In addition, we evaluated the prognostic value of the combination of these three proteins for TNBC. Based on the expression levels of these three proteins, clinical patients were classified into high-risk or low-risk groups. We assigned patients with low levels of expression of more than two of the three proteins to the high-risk group, and the remaining patients were assigned to the low-risk group. By comparing the survival curves of the two groups, we found that the survival time of patients in the high-risk group was much shorter than the survival time of patients in the low-risk group (Figure 8B). This result suggest that the combination of these three proteins can also be used as a promising predictor of prognosis for TNBC patients. The consistency of a comprehensive analysis of these three proteins suggests that the combination of identified candidate genes is reliable for the prognostic assessment of TNBC. 

## 4. Discussion

BC is currently the primary cause of cancer-related deaths in women, having surpassed lung cancer to become the most prevalent form of malignant tumor globally [23]. TNBC is a particularly aggressive subtype of breast cancer that is characterized by a high risk of both local recurrence and distant metastasis. Unfortunately, it also currently lacks effective targeted therapy options, making it a significant challenge in breast cancer treatment [24,25]. This study identified the M1/M2 macrophage ratio as a novel prognostic marker for TNBC, using a Pearson correlation analysis. The related genes were grouped into samples using univariate Cox regression dimension reduction. To validate our riskScore model, we conducted a comprehensive annotation of the TME, clinical features, associated pathways, genomics, and immunotherapy response, based on the differences observed between the two clusters.

TAMs comprise more than 50% of the immune cells that infiltrate the TME and play a significant role in promoting tumor growth and metastasis. Notably, TAMs can be classified into two subtypes, M1 macrophages and M2 macrophages, based on their functions. In general, their opposite effect results in the suppression of anti-tumor immune responses, and this effect is more pronounced in TNBC. This is because TNBC secretes more granulocyte colony-stimulating factor (G-CSF) than other BC subtypes, thus promoting the transition from M1 to M2 macrophages [26]. In addition, CD163 and CD204 (molecular markers of M2 macrophages) in M2 macrophages are closely associated with histologically aggressive characteristics and a worse TNBC prognosis [21,27,28].TAMs can also reduce the effector function of TILs and promote Tregs to promote tumor growth and progression. In conclusion, TAMs can directly or indirectly mediate the tumor growth and invasion of TNBC. 

To investigate and validate the predictive value of M1/M2 macrophages in TNBC, the three most significant genes were identified in the DEGs. A riskScore model was developed using these three genes. The M1/M2 macrophage-related riskScore was highly efficient in predicting the probability of patient survival at 3 and 5 years, and the clinical and survival differences between patients in the high and low riskScore groups were significant. Most importantly, almost all immune cell infiltration decreased with increasing riskScore values except for fibroblasts and M2macrophages, reflecting a synergistic effect between fibroblasts and M2 macrophages. In addition, patients with a high riskScore value showed higher degrees of negative correlation with the GO pathways related to macrophage activation, macrophage migration, and G-CSF secretion. Classical IC molecules such as PD-1, PD-L1, ICOS, and CTLA4 showed low levels of expression in the high riskScore group. Furthermore, patients’ ESTIMATE and ImmuneScore values decreased with increasing riskScore values, suggesting that a poorer antitumor TME is associated with a higher riskScore value.

Currently, modulating TAMs into M1 macrophages is a potential strategy for improving the prognoses of TNBC patients. The selective Class IIa Histone Deacetylase 2 (HDAC2) inhibitor TMP195 has been shown to convert TAMs to an M1-like macrophage phenotype and therefore has demonstrated impressive efficacy, especially in combination with paclitaxel [29]. Considering the intrinsic nature of the riskScore as a reflection of M1/M2 macrophages at the genetic level, the riskScore could serve as a very prospective indicator of the efficacy of HDAC2 inhibitor therapy. If patients achieve lower riskScore values after treatment, they can be considered to have had good responses to HDAC2 inhibitors.

The inhibition of TAMs recruitment has emerged as a promising approach for improving prognoses in TNBC patients. Recent studies have shown that blocking CSF-CSF receptor (CSF1R) signaling can reduce TAMs recruitment, enhance the effect of paclitaxel, and prolong the survival time of mice [26,30]. In addition, it increases cytotoxic T lymphocyte (CTLs) infiltration and decreases vascular density by reducing VEGF mRNA expression [26]. Based on the negative impact of riskScore on CSF production, the VEGF signaling pathway, and CTL migration, riskScore may serve as a valuable marker for selecting TNBC patients who can benefit from anti-CSF-CSFR agents such as Emactuzumab. A high pretreatment riskScore value indicates resistance to the regimen, and these patients may not benefit from such treatments. Surgery has also been found to affect the number of TAMs, the level of IL-6, and the level of chemokine (C-C motif) ligand 5 (CCL5), all of which are associated with angiogenesis, cell migration, and tumor cell invasion in TNBC [31,32]. Thus, riskScore can be utilized to assess the risk of postoperative recurrence. In summary, targeting the recruitment of TAMs by blocking CSF-CSF receptor signaling is a promising approach for improving prognoses in TNBC patients. The riskScore may serve as a valuable marker for selecting patients who can benefit from such treatments, and it can also be used to assess the risk of postoperative recurrence.

Genomic variants associated with M1/M2 macrophages were investigated. The KMT2D mutation provided better survival outcomes for TNBC patients and increased the level of infiltration of immune cells, including macrophages [19]. Nevertheless, this study did not reveal any significant difference in KMT2D missense mutations between the two riskScore groups. This may be attributed to the limited number of TNBC samples in the TCGA database or the potential interaction of KMT2D with other genes in TNBC. Nonetheless, KMT2D deficiency has been shown to increase tumor sensitivity to various types of immune checkpoint inhibitors [20]. In summary, mutational differences in KMT2D and other related genes between high and low riskScore groups must be confirmed by further studies.

Anti-PD-1 immunotherapy has been considered a potential therapeutic option for advanced TNBC, but clinical trials have not yet shown satisfactory results [33]. Based on the IMvigor210 cohort, our study showed that patients with higher riskScore values had lower survival probability curves and that patients who achieved CR or SD tended to have lower riskScore values, indicating that they responded well to respond to ICIs. For anti-TAM treatments, such as anti-CSF-CSR or HDAC2, although we did not analyze them in the corresponding cohorts, a GSEA showed that the mechanisms associated with these strategies were closely related to riskScore. Thus, riskScore may be a potential indicator and marker for patients receiving ICI treatments.

## 5. Conclusions

In summary, our analysis establishes an M1/M2 riskScore model consisting of three specific genes and evaluates its predictive and prognostic value for TNBC. Our findings greatly support the moderating role of the M1/M2 macrophage ratio in TNBC progression and confirm the role of M1/M2 macrophage ratio in predicting the probability of patient survival and the efficacy of immunotherapy.

Our findings should be followed by single-cell RNA sequencing to determine the expression of genes on different cells and, considering the importance of the spatial distribution of immune cells in the TME, it is necessary to apply spatial transcriptomics to assess the infiltration of immune cells according to their location. In addition, a more detailed molecular mechanistic analysis of these genes will be performed, applying chromatin immunoprecipitation to explore the regulatory mechanisms of the genes, as well as their downstream regulatory genes to elucidate their roles in promoting TNBC progression and metastasis. Since these genes are overexpressed in tumor cells and tumor tissues, we hypothesize that changes in the expression of these genes may also be present in circulating tumor cells (CTCs). Detection of the expression of these genes in CTCs is expected to provide an early diagnosis of TNBC. In addition, the deeper mechanisms, such as the regulation of the differentiation of M0 to M1 or M2 macrophages in the TNBC TME, still require further experimental validation. Finally, the TME of uroepithelial carcinoma may be different from that of TNBC, and the absence of a TNBC cohort cannot accurately predict TNBC response to immunotherapy. Therefore, a TNBC cohort is needed to validate the effectiveness of our riskScore model in predicting immunotherapy response.

## Figures and Tables

**Figure 1 medicina-59-01285-f001:**
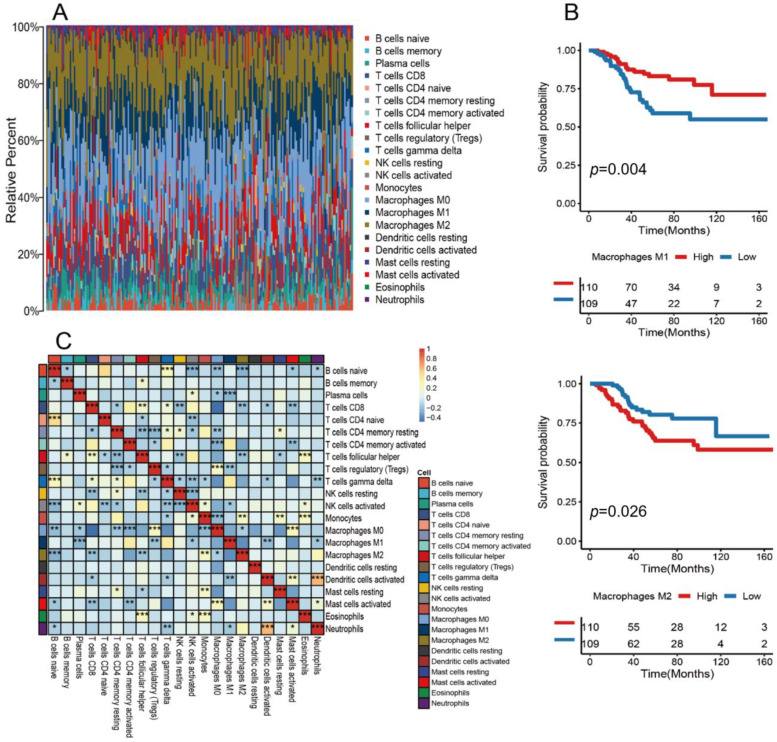
Landscape and macrophage survival analysis of the TME in TNBC: (**A**) cellular composition of the TME. Each cell on the horizontal axis represents one of 219 samples from the TCGA and GEO datasets. The vertical axis shows the composition of immune cells. The different colors represent the corresponding types of immune cells. The column length of a particular color indicates the relative number of immune cells; (**B**) Kaplan–Meier curves for overall survival (OS) were plotted for 219 TNBC patients divided into two groups based on the median numbers of M2 macrophages and M1 macrophages; (**C**) correlation heat map of 22 classes of immune cells coexisting or mutually exclusive (* *p*-value < 0.05; ** *p*-value < 0.01; *** *p*-value < 0.001).

**Figure 2 medicina-59-01285-f002:**
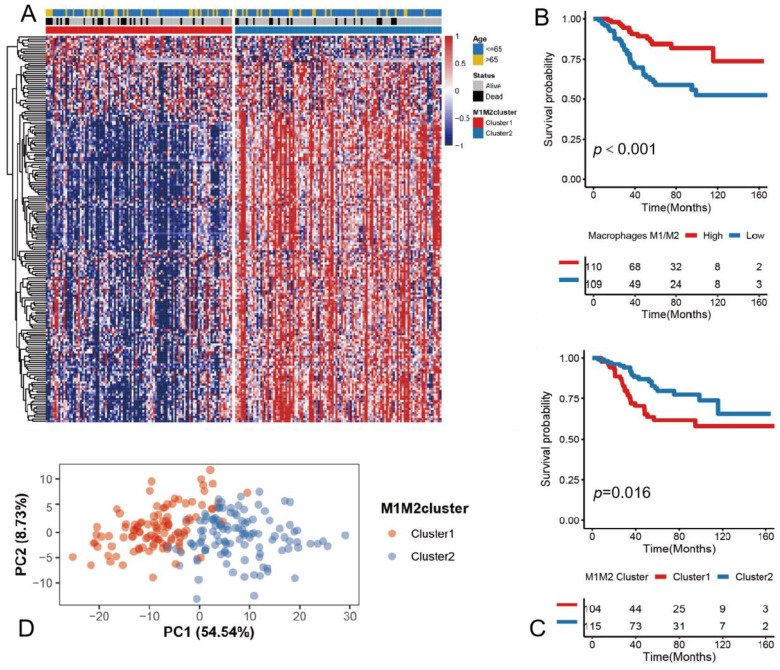
Classification of samples into 2 clusters based on genes associated with M1/M2 macrophages: (**A**) heatmap of M1M2 macrophage–related gene expression profiles. Clustering and other clinical information are shown on the right side as patient annotations. Rows indicate associated genes and columns indicate samples; (**B**) Kaplan–Meier survival analysis of the two clusters grouped according to M1/M2 macrophages; (**C**) Kaplan–Meier survival analysis of two clusters grouped according to M1/M2 macrophages related genes; (**D**) PCA analysis shows good separation between M1M2cluster1 and M1M2cluster2.

**Figure 3 medicina-59-01285-f003:**
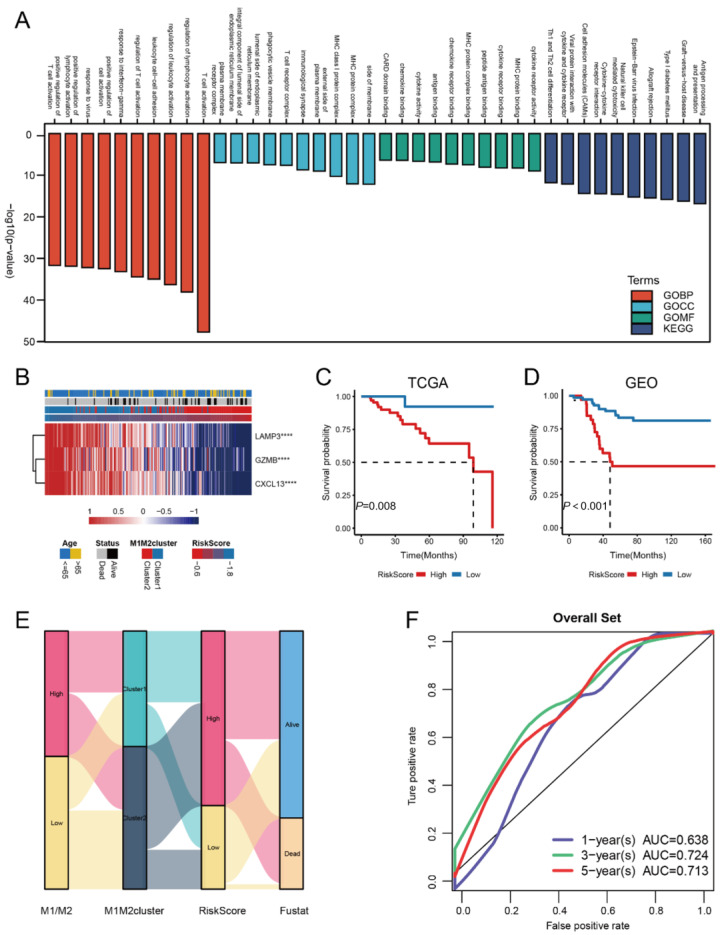
(**A**) riskScore model construction and its validity, GO, and a KEGG enrichment analysis of 109 DEGs between M1M2cluster1 and 2; (**B**) LAMP3, GZMB, and CXCL13 were used to construct the riskScore model. The riskScore value increases from left to right, as shown by the depth of colore (**** *p*-value < 0.0001); (**C**) Kaplan–Meier survival analysis of two clusters in TCGA using the riskScore method; (**D**) Kaplan–Meier survival analysis of two clusters in GEO using riskScore; (**E**) alluvial diagram showing the correlation between M1/M2, M1M2 clusters, riskScore, and patient survival; (**F**) ROC (Receiver operating characteristic) curves measuring the sensitivity of riskScore when predicting patient survival status at 1, 3, and 5 years. The AUC (Areas under the ROC curves) were 0.638, 0.724, and 0.713, respectively.

**Figure 4 medicina-59-01285-f004:**
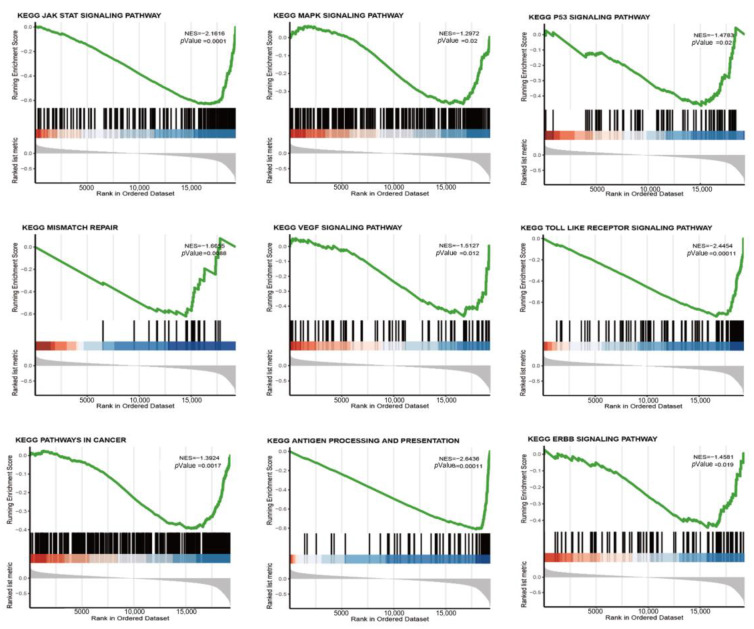
The association of riskScore with important pathways in the KEGG was assessed via a GSEA. The results of the GSEA revealed significant correlations between riskScore and several pathways, including the mismatch repair, JAK/STAT3 signaling pathway, VEGF signaling pathway, antigen processing presentation, ERBB signaling pathway, and P53 signaling pathway. The statistical significance of each association was calculated using log-rank tests.

**Figure 5 medicina-59-01285-f005:**
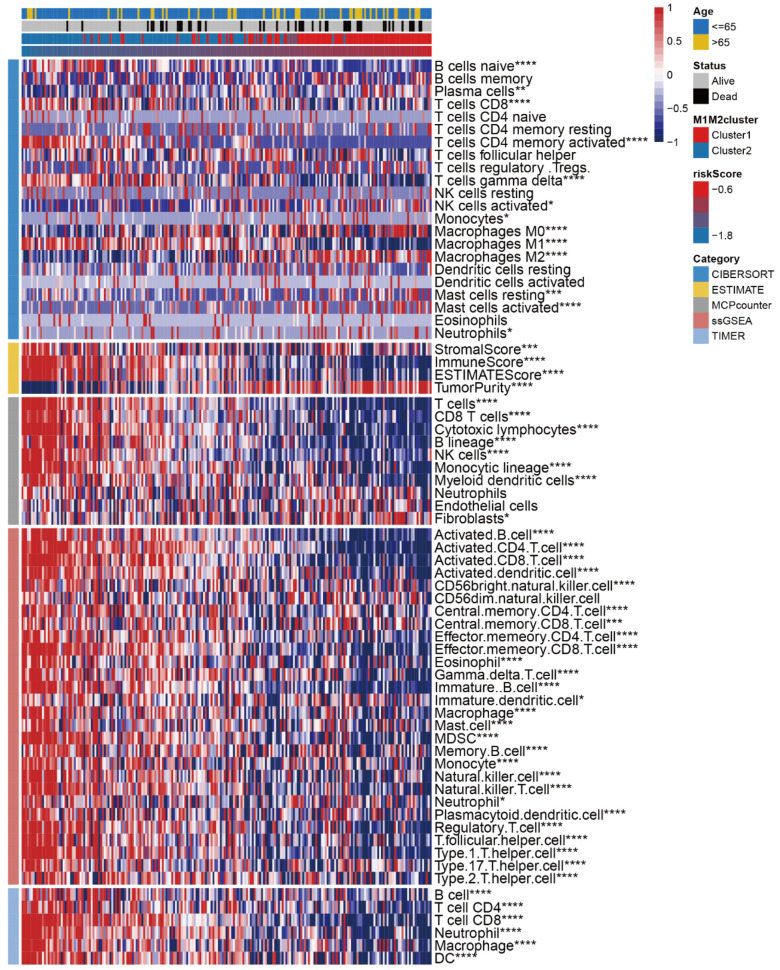
Immunological characteristics of the riskScore model. Pearson correlation analysis analyzed riskScore at the level of 64 cell types (* *p*-value < 0.05; ** *p*-value < 0.01; *** *p*-value < 0.001; **** *p*-value < 0.0001; NS, not statistically significant).

**Figure 6 medicina-59-01285-f006:**
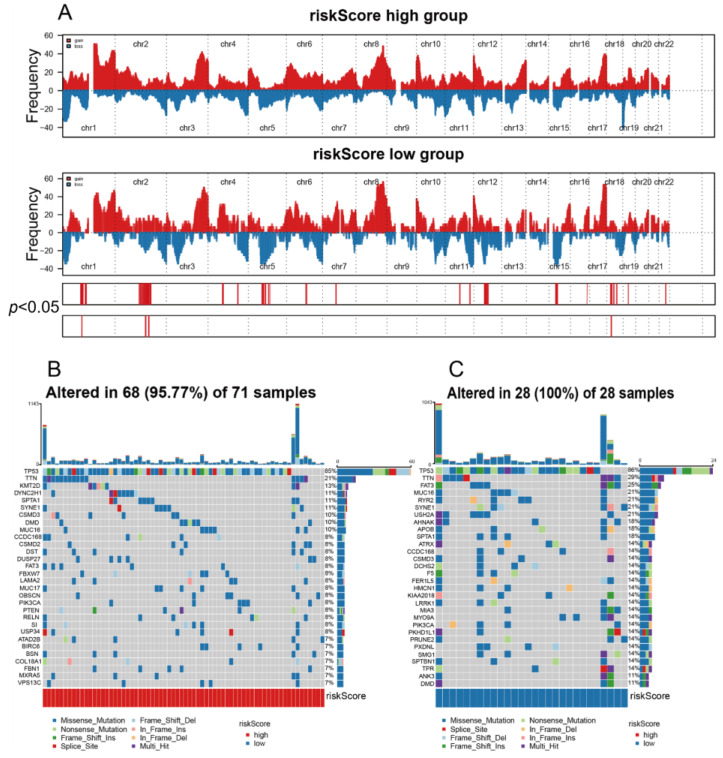
Characterization of the two clusters: (**A**) distribution of gain or loss of function mutations in the 22 human chromosomes of the two clusters. Gene amplifications are indicated in red. Gene deletions are marked in blue; (**B**) list of the most frequently altered genes in M1M2cluster 1; (**C**) list of the most frequently altered genes in M1M2cluster 2. A total of 8 mutation types occurred.

**Figure 7 medicina-59-01285-f007:**
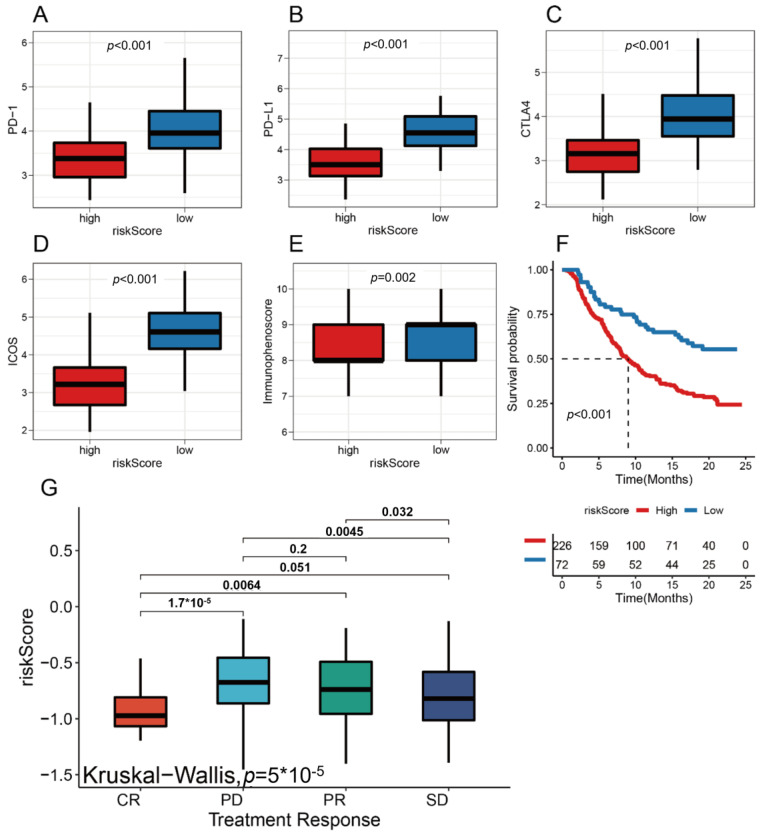
Association of immunotherapy-related genes with prognosis and riskScore: (**A**) box line plot of PD-1 levels in high and low riskScore groups; (**B**) box line plots of PD-L1 levels in high and low riskScore groups; (**C**) box line plots of CTLA4 levels in high and low riskScore groups; (**D**) box line plots of ICOS levels in high and low riskScore groups; (**E**) ImmunophenotypeScore in the high and low riskScore groups; (**F**) Kaplan–Meier survival analysis of the two groups treated with ICIs; (**G**) different ICI treatment responses and their corresponding riskScore values.

**Figure 8 medicina-59-01285-f008:**
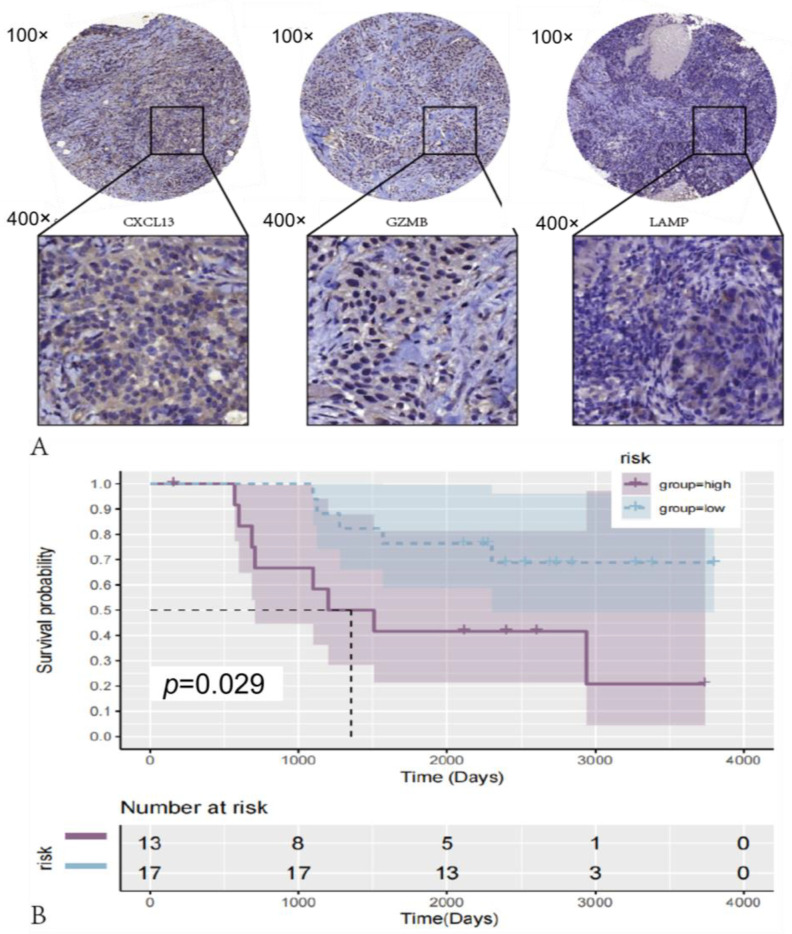
Differentially expressed proteins in human TNBC tissue: (**A**) the protein expression of LAMP, GZMB, and CXCL13 in clinical human TNBC tissue was detected via IHC. Representative photos are shown (100× and 400×). Scale bar = 100 μm. (**B**) Based on the expression scores of differentially expressed proteins, survival curves of patients in high-risk and low-risk groups. Patients with low levels of expression of two or more of these three proteins were defined as belonging to the high-risk group, and the others belonged to the low-risk group.

## Data Availability

The data used to support the findings of this study are available from the corresponding author upon request.

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
