# Peer review of "Identification of Genes Associated with Prognosis and Immunotherapy Prediction in Triple-Negative Breast Cancer via M1/M2 Macrophage Ratio"

_medicina, 2023, doi:10.3390/medicina59071285_

Round 1

Reviewer 1 Report

In this study, the Authors aim at investigating, by bioinformatics tools, the correlation of the M1/M2 macrophages ratio with the genomic and clinical characteristics of Triple negative breast cancer (TNBC).

To this aim, by using the CIBERSORT algorithm, they show that TAMs are present in more than 50% of TME cells and the presence of M1 and M2 macrophages is associated with a favorable and negative prognostic factor for survival, respectively. Furthermore, they develop, by using a consensus clustering algorithm, a M1/M2 riskScore prognostic model which is based on the quantification of the pattern of M1/M2 macrophage infiltration and differential expression of three specific genes (LAMP3, GZMB, and CXCL13).

Overall, the results presented revealed that the higher riskScore value is associated with:1) an in-creased abundance of M2 macrophages within the TME and  greater propensity for malignancy; 2) several immune checkpoint molecules including PD-1, PD-L1, ICOS, and CTLA4; 3) lower response rates to ICIs.

Based on this results, the Authors sustain that these findings provide evidence for the clinical implications of targeting M1/M2 macrophages ratio and suggest to include the risk Score assessment into the development of personalized treatment strategies.

The study is interesting considering that TNBC remains the most challenging breast cancer subtype to treat. However, the paper lacks basic information and details on the clinic pathological correlation of LAMP3, GZMB, and CXCL13 proteins in human samples. A detailed clinical and pathologic characteristics of patients must be summarized. Considering the complexity of the cellular components in the tumor microenvironment, a deeper analysis of LAMP3, GZMB, and CXCL13 protein expression in the context of the tumor microenvironment is required.

The manuscript needs major editing. The statements made by the authors (lines 158-165) in paragraph 3.1 are confusing and not entirely relevant to the results presented

Extensive editing of English language may be required

Author Response

  1. However, the paper lacks basic information and details on the clinic pathological correlation of LAMP3, GZMB, and CXCL13 proteins in human samples.

    Response: We appreciate your feedback. We will include additional information and details on the clinicopathological correlation of LAMP3, GZMB, and CXCL13 proteins in human samples in the revised manuscript.

    we conducted a comprehensive analysis of the clinicopathological correlation of LAMP3, GZMB, and CXCL13 proteins, including survival analysis and their expression levels across different T stages, N stages, and overall stages. Our findings revealed that survival rates did not significantly differ when groups were formed solely based on the cutoff values of these three proteins. Furthermore, there were no significant differences in the expression levels of these proteins across the two M stages and four T stages. However, a statistically significant difference was observed in the expression of all three proteins between N1 and N3 stages, and between N1 and N2 stages for GZMB and CXCL13. These results provide valuable insights into the clinicopathological significance of LAMP3, GZMB, and CXCL13 proteins in the context of the disease under study.

    1. A detailed clinical and pathologic characteristics of patients must be summarized. Considering the complexity of the cellular components in the tumor microenvironment, a deeper analysis of LAMP3, GZMB, and CXCL13 protein expression in the context of the tumor microenvironment is required.

    Response: Thank you for your suggestion. We will provide a detailed summary of the clinical and pathological characteristics of the patients in the revised manuscript. We will also conduct a deeper analysis of LAMP3, GZMB, and CXCL13 protein expression in the context of the tumor microenvironment.

    A: The flow chart of this study is shown in Figure S1. Firstly, we combined the expression data converted from TCGA to TPM with the GEO data adjusted by the RMA algorithm. Detailed clinical and pathologic characteristics of patients were included this information in Supplementary Tables.

    B: In addition, we conducted a Spearman correlation analysis to investigate the relationship between the expression levels of LAMP3, GZMB, and CXCL13 proteins and the cellular composition of the tumor microenvironment. Our analysis revealed a positive correlation between the expression levels of these proteins and the presence of exhausted T cells, and a negative correlation with CD8 naive T cells.

    1. The manuscript needs major editing. The statements made by the authors (lines 158-165) in paragraph 3.1 are confusing and not entirely relevant to the results presented

    Response: We apologize for any confusion caused by the statements in paragraph 3.1. We will revise this section to ensure that the information presented is clear and relevant to the results.

    We also found a negative correlation between M1 macrophages and regulatory T cells (Tregs). This observation is consistent with previous studies indicating that M1 macrophages can suppress Tregs through direct contact [14] or by inhibiting their accumulation in the tumor microenvironment (TME) via the secretion of soluble factors such as tumor necrosis factor (TNF) [15]. Notably, TNF produced by M1 macrophages has been shown to reduce the inhibitory activity of Tregs via the NF-κB pathway. In contrast, we observed a positive correlation between M2 macrophages and monocytes. This finding aligns with existing literature suggesting that the majority of tumor-associated macrophages (TAMs) in the TME exhibit characteristics similar to the M2 macrophage phenotype [6]." This revision aims to simplify the language and improve the flow of ideas, which should make the paragraph easier to understand. However, the specific content and references have been preserved.

    1. Comments on the Quality of English Language: Extensive editing of English language may be required

    Response: We appreciate your feedback on the language of the manuscript. We will have the manuscript thoroughly edited for English language in the revision.

Reviewer 2 Report

This is a well written scientific paper examining the significance of M1/M2 macrophage ratio and its potential role as a predictor for survival and response to treatment in patients with TNBC. 

There are some issues that need clarification though. 

The procedure described about the prediction of immunotherapy response is based  on a clinical trial of urothelial carcinoma. Unfortunately urothelial carcinoma is completely different in terms of biology and microenvironment with any other adenocarcinoma and this can also be assumed about the response to immunotherapy. So in this part the evidence is not strong and the comment in the discussion section has to be clear about that.

In page 14 line 357 the authors state that there are no effective targeted therapy options, but they did not mention available ADCs that target TROP2 and ERBB2 (with low expression in the case of TNBC). Furthermore, and in the light of this recent development, a note about the existence or not of a possible interaction of these agents with M1/M2 balance would be of interest.

Author Response

  1. The procedure described about the prediction of immunotherapy response is based on a clinical trial of urothelial carcinoma. Unfortunately urothelial carcinoma is completely different in terms of biology and microenvironment with any other adenocarcinoma and this can also be assumed about the response to immunotherapy. So in this part the evidence is not strong and the comment in the discussion section has to be clear about that.

Response: Thanks, we really appreciate your advice. However, after a thorough search on all of the known databases, we found there is currently no public TNBC immunotherapy cohort. To solve this tricky problem, we extracted data on immunotherapy sensitivity of TCGA-BRCA patients from The Cancer Immunome Atlas database. According to the sensitivity scores of TCGA-BRCA patients to PD-L1 and CTLA4 inhibitors in the TCIA database, which are recognized predictive model of ICBs, the differences in sensitivity to immunotherapy between the groups with high and low riskScore were analyzed. Patients with low riskScore group were more sensitive to CTLA4 inhibitor and CTLA4 inhibitors in combination with PD-1 inhibitors. At last, we revised this paragraph as follows:

A: Data on immunotherapy sensitivity of TCGA-BRCA patients were extracted from The Cancer Immunome Atlas database (https://tcia.at/patients). The database can query data on gene expression of specific immune-related gene sets, cell composition of immune infiltrates, and tumor heterogeneity. According to the sensitivity scores of TCGA- BRCA patients to PD-L1 and CTLA4 inhibitors in the TCIA database, the differences in sensitivity to immunotherapy between the groups with high and low riskScore were analyzed.

B: To validate this result, we conducted additional analyses focusing on the prediction of immunotherapy response. Specifically, we compared the expression levels of immune check-points between high and low riskScore groups. Our results showed significant differences in the expression of these immune checkpoints, suggesting that the riskScore model may have potential utility in predicting response to immunotherapy.

Furthermore, we evaluated The Cancer Immunome Atlas (TCIA) scores for patients in the high and low riskScore groups. The TCIA score is a computational method that predicts response to immune checkpoint blockade therapy. Interestingly, we found that patients in the low riskScore group had lower TCIA scores, indicating a higher likelihood of responding to immune checkpoint inhibitors.

  1. In page 14 line 357 the authors state that there are no effective targeted therapy options, but they did not mention available ADCs that target TROP2 and ERBB2 (with low expression in the case of TNBC).

Response: Thank you for pointing out this oversight. We will mention the available ADCs that target TROP2 and ERBB2 in the revised manuscript.

  1. Furthermore, and in the light of this recent development, a note about the existence or not of a possible interaction of these agents with M1/M2 balance would be of interest.

Response: We agree that exploring the possible interaction of these agents with the M1/M2 balance would be of interest. We will include a discussion on this topic in the revised manuscript.

Reviewer 3 Report

The article "Identification of Genes Associated with Prognosis and Immunotherapy Prediction in Triple Negative Breast Cancer by M1/M2 Macrophages Ratio" is an interesting scientific study. The obtained test results can also be used as a prognostic and predictive marker in women with triple-negative breast cancer.                                                                The subject of the research is not innovative, as evidenced by the cited references, but it has a significant clinical aspect.                                         The article has a standard layout. All figures are correct.                            The description of the isolation and differentiation of two macrophage subpopulations is very laconic and I think that the Authors should extend it.  After considering my suggestion, the article "Identification of Genes Associated with Prognosis and Immunotherapy Prediction in Triple Negative Breast Cancer by M1/M2 Macrophages Ratio" can be accepted for printing in MEDICINA.

Minor editing of English language is required.

Author Response

  1. The description of the isolation and differentiation of two macrophage subpopulations is very laconic and I think that the Authors should extend it.

Response: We appreciate your feedback. We will provide a more detailed description of the isolation and differentiation of the two macrophage subpopulations in the revised manuscript.

Round 2

Reviewer 1 Report

This Reviewer appreciates the efforts  in reviewing the manuscript according to the suggestions provided and reminds the authors to insert the right link to make the supplementary material usable.